# Evolving Patterns of Nutritional Deficiencies Burden in Low- and Middle-Income Countries: Findings from the 2019 Global Burden of Disease Study

**DOI:** 10.3390/nu14050931

**Published:** 2022-02-22

**Authors:** Jingjing Liu, Xinye Qi, Xing Wang, Yinghua Qin, Shengchao Jiang, Liyuan Han, Zheng Kang, Linghan Shan, Libo Liang, Qunhong Wu

**Affiliations:** 1Department of Health Policy, Health Management College, Harbin Medical University, Harbin 150086, China; 15001290091@163.com (J.L.); qixinye1992@163.com (X.Q.); qinyinghua250@126.com (Y.Q.); jshengchao1@163.com (S.J.); kangzheng0086@126.com (Z.K.); linghanshan@126.com (L.S.); 2Department of Social Medicine, School of Public Health, Harbin Medical University, 157 Baojian Road, Nangang District, Harbin 150086, China; 3The Fourth Affiliated Hospital, School of Medicine, Zhejiang University, Yiwu 322000, China; wx1454164617@163.com; 4Hwa Mei Hospital, University of Chinese Academy of Sciences, Ningbo 315000, China; hanliyuan@ucas.ac.cn

**Keywords:** nutritional deficiencies, malnutrition, incidence, disability-adjusted life years

## Abstract

Low- and middle-income countries (LMICs) suffered the most from nutritional deficiencies (NDs). Although decades of efforts have reduced it, little is known about the changing trajectory of ND burden in LMICs. By extracting data of the Global Burden of Diseases, Injuries, and Risk Factors Study 2019, we calculated indicators of incidence and disability-adjusted life years (DALYs) to measure the burden of NDs and its main subcategories in LMICs, including protein-energy malnutrition, iodine deficiency, vitamin A deficiency, dietary iron deficiency, and other nutritional deficiencies by sex, age and spatial patterns. In LMICs, ND incidence still increased in the age group 15+ born before 2005, especially in males. The effort of reducing the DALYs of NDs has generated a strong decline in per age group. In the main subcategories of NDs, protein-energy malnutrition incidence in males age 45+ born before 1970 still increased. Despite vitamin A deficiency incidence and dietary iron deficiency, DALYs strongly experienced decreases over three decades while still remaining at the heaviest level in 2019, especially in females and children under 5 years. The top largest tendency estimates occurred in Mali’ females and Bhutan’ males. Zimbabwe was the only country with increased DALYs rate tendency in both sexes.

## 1. Introduction

Despite progress in achieving the United Nations Decade of Action on Nutrition 2016–2025 and the Sustainable Development Goal to “eliminate all forms of malnutrition”, nutritional deficiencies (NDs) remain widespread in low- and middle-income countries (LMICs) [1]. Vitamin A deficiencies and insufficient dietary zinc intake are common in South Asia and sub-Saharan Africa [2]. There are 13 countries where the prevalence of iodine deficiencies in preschool children is more than 50%, and the prevalence of anemia caused by iron deficiencies exceeds 40% in Africa [3]. Severe iodine deficiencies in Ethiopian female causes 50,000 stillbirths each year [4]. NDs primarily hamper national growth and prevent economic stability in LMICs due to reduced productivity and cognition, increased susceptibility to infectious and chronic diseases, and elevated treatment costs [5]. However, in the context of the coronavirus disease 2019 (COVID-19) pandemic, protein, vitamin A, or zinc, etc., have been proved to possess associations with increased infection risks [6].

There is no universal agreement on the definition and clinical assessment of NDs, but it generally includes two major forms: protein-energy malnutrition and micronutrient deficiencies [7]. Epidemiological evidence of NDs in general practice is rare, and published data are disputable [8]. Moreover, previous studies have failed to consider sex as a confounder. Due to a combination of biological and social factors, sex-related differences are striking in energy metabolism, nutritional requirements, physical performance, appetite regulation, nutritional strategy choice, and nutritional behavior [9]. Driven by nutritional transition, sociocultural beliefs, livelihood models, occupations, everyday diet, and physical activity patterns have significantly changed among both sexes in LMICs [10].

We aimed to produce an assessment of the burden, trends, and inequalities of NDs in LMICs by use of a comprehensive approach, which included sex-specific secular trends and birth cohort trend analysis by conducting a secondary analysis of the Global Burden of Diseases, Injuries, and Risk Factors Study 2019 (GBD 2019), and a classification of countries based on their evolving pattern of NDs burden to identify where NDs remained a public health issue.

## 2. Materials and Methods

### 2.1. GBD Data Set

We obtained nutritional deficiencies (NDs) incidence and disability-adjusted life years (DALYs) data from GBD 2019. GBD 2019 data used all available sources of epidemiological data abstracted from censuses, household surveys, civil registration and vital statistics, disease registries, health service use, air pollution monitors, satellite images, and disease notifications at the time and improved standardized methods to estimate the incidence, prevalence, mortality, years of life lost (YLLs), years lived with disability (YLDs), and DALYs due to 369 diseases and injuries in 204 countries and territories [11]. A detailed description of the metrics, data sources, and statistical modeling used in the GBD 2019 was reported elsewhere [12]. GBD 2019 was in accordance with the Guidelines for Accurate and Transparent Health Estimates Reporting statement [13].

NDs were identified based on the 10th revision of the International Classification of Diseases and Injuries (ICD-10), coded D50−D53.9, E00−E02, E40−E46.9, E50−E61.9, E63−E64.9, Z13.2−Z13.3. Its main subcategories included protein-energy malnutri-tion (coded E40−E46.9, E64.0), iodine deficiencies (coded E00−E02), vitamin A deficiencies (coded E50−E50.9, E64.1), dietary iron deficiencies (coded D50−D50.9), and other nutritional deficiencies (coded D51−D53.9, E51−E61.9, E63−E64, E64.2−E64.9).

### 2.2. Gender Development Index Set

In this study, we explored the relationship between sex-specific NDs incidence and DALYs burden with Gender Development Index (GDI). GDI is a direct measure of sex gap in human development achievements. The GDI is a direct measure of the sex gap in human development achievements by accounting for sex disparities in health, knowledge, and living standards using the same methodology as in the Human Development Index (HDI). GDI is the ratio of HDIs calculated separately for female and male and shows the female HDI as a percentage of the male HDI. The GDI score is closer to 1, indicating that the sex difference is smaller. Data were acquired from the United Nations Development Programme (UNDP; http://hdr.undp.org/en/data, accessed on 1 June 2021).

### 2.3. Statistical Analysis

One-hundred and thirty-four countries of two hundred and four GBD countries showed per capita national income less than USD 12,535 $ in 2019, which is the boundary with high income defined by the World Bank classification system [14], and these 134 low- and middle-income countries were included in our study. We firstly performed descriptive analysis to characterise the sex-specific burden of NDs and its main subcategories in overall LMICs across the number of cases and age-standardized rate (per 100,000). We calculated the overall calendar-year cases by summing the cases from 134 countries and calculated the overall calendar-year rate using the number of new cases as the numerator and summing the mid-year population size in 134 countries as the denominator. Consistent with the methods used in the initial GBD study. The world standard population was used to calculate the age-standardized rate [15].

We evaluated sex-specific incidence and DALYs temporal trends among all ages for NDs using log-linear regression models and further described age and sex-specific trends of high-risk subcategories from 1990 to 2019 by the same method. The trends results were presented as the estimated annual percentage change (EAPC) and the corresponding 95% CI to measure the average increase in the age-standardized incidence rate (ASIR) and age-standardized incidence rate (ASDR) over 30 years [15].

Changes with increased incidence risk born at different times were investigated graphically using birth cohort curves for sex separately. We then explored birth cohort effects using age-period–cohort models. These models used 20 5-year age groups (1 to 5, …, 95+) and 6 5-year calendar periods (1990 to 1994, …, 2014 to 2019). Detailed description of age-period–cohort methods is available in the Supplementary Methods of Best et al. [16]. From these models, we calculated cohort-specific incidence rate ratios for each birth cohort stratified by sex, with the 1955 cohort used as the reference (relative risk of 1.0). The analysis was conducted using age-period–cohort model fitting software developed by the National Cancer Center. We further investigated annual percent change (APC) across the cohort-specific incidence rate ratios by fitting Joinpoint log-linear regression models [17]. The positive slopes in APC for Joinpoint segments indicate that rate ratios increased, whereas negative slopes indicate a decline in rate ratios. Moreover, the model-selected Joinpoints indicate a change in the rate of increase/decrease (*p* < 0.05). The analysis was conducted using the Joinpoint Regression Program, Version 4.8.0.1—April 2020; Statistical Methodology and Applications Branch, Surveil-lance Research Program, National Cancer Institute.

We finally analyzed the relations between sex-specific burden and Gender Development Index (GDI) using Pearson’s correlation analysis, and all analyses used SAS 9.4 statistical software (SAS Institute Inc., Cary, NC, USA) and R version 3.3.

## 3. Results

### 3.1. The Evolving Pattern of the Contribution in LMICs to Global NDs Burden

The proportion of LMICs incidence number in the global NDs burden increased from 1990 to 2019, with the ratio from 55.2% to 70.1% in females and 59.3% to 73.2% in males (Figure 1A). The absolute incidence number of NDs in LMICs increased in both sexes from 1990 to 2019, with females increasing 66.0% (31.0 million in 1990 to 51.4 million in 2019) and males increasing 74.1% (from 37.4 million to 65.1 million) (Appendix A). More than 80% of global DALYs cases caused by NDs occurred in LMICs, with the proportion of 85.5% females and 81.9% males in 2019, increased by 13.9% and 16.1%, respectively, over 1990 (Figure 1B). The DALYs absolute number of NDs in LMICs decreased in both sexes from 1990 to 2019, with females decreasing 26.8% (from 33.2 million to 24.3 million) and males decreasing 36.0% (from 27.3 million to 17.5 million) (Appendix A).

With the exception of protein-energy malnutrition, the other main subcategories of NDs in LMICs contributed over 80% incidence and DALYs cases to global burden across 30 years. However, the incidence and DALYs absolute number of protein-energy malnutrition in LMICs showed the fastest increased contribution to global cases, in which incidence cases increased from 51.7% to 68.1% in females and 57.3% to 72.4% in males; DALYs cases increased from 58.5% to 65.5 % in females and 52.5% to 65.4% in males (Appendix A).

### 3.2. Temporal Trends and Main Subcategories Differences of NDs in LMICs

During the 30-year time frame, the overall age-standardized incidence rate (ASIR) of NDs was stable for females; however, it increased in males from 1686.4 per 100,000 in 1990 to 2086.6 per 100,000 in 2019, with an annual average of 0.7%. The age-standardized DALYs rate (ASDR) due to NDs decreased significantly in females (−2.3% per year, from 672.1 to 174.7 per 100,000) and males (−2.6% per year, from 583.4 to 169.8 per 100,000) during 1990 and 2019 (Table 1 and Appendix A).

Figure 2 and Appendix A showed the sex-specific age-standardized rate of main subcategories in LMICs by calendar year, 1990–2019. Of NDs’ main subcategories, vitamin A deficiencies contributed to the largest ASIR over 30 periods. The largest ASDR caused by protein-energy malnutrition shifted to dietary iron deficiencies in females since 1992; however, the pattern has shifted among males since 2003.

The ASIR of vitamin A deficiencies showed the fastest declining subcategory in females (−3.4% per year, from 13,760.7 to 6238.7 per 100,000) and males (−2.7% per year, from 22,282.0 to 8325.3 per 100,000) from 1990 to 2019. Notably, the ASIR of protein-energy malnutrition among males observed the only increased subcategory with an annual average of 0.8%. All subcategories were observed with decreased ASDR, especially protein-energy malnutrition in both females (−4.6% per year, from 672.1 to 174.7 per 100,000) and males (−4.3% per year, from 583.4 to 169.8 per 100,000) over the study period. Of note, dietary iron deficiencies showed the slowest decline over females (−0.7% per year, from 639.2 to 518.6 per 100,000) and males (−0.9% per year, from 416.3 to 316.5 per 100,000).

### 3.3. Age-Sex Specific Trends of NDs and Its High-Risk Subcategories in LMICs

Figure 3, Appendix A provide detailed information of age-sex specific trends of overall NDs and its high-risk subcategories (protein-energy malnutrition with increasing risk; vitamin A deficiency with the heaviest ASIR; and dietary iron deficiency with the heaviest ASDR). The ASIR and ASDR of NDs and its high-risk subcategories had always been ahead in aged 1 to 4 than older age groups during 1990 to 2019.

Except for ages 1 to 4 and 5 to 14, which are kept generally stable, other age groups showed increased ASIR of NDs. Females aged 15 to 44 (0.8% per year, from 1016.0 to 1338.2 per 100,000) and 75+ years (0.8% per year, from 1061.7 to 1538.0 per 100,000) were observed the fastest increased tendency. In males, ASIR increased strongly in age 70+ (1.1% per year, from 1511.5 to 2199.2 per 100,000) and age 60 to 74 (1.1% per year, from 1417.3 to 2090.8 per 100,000) from 1990 to 2019 (Figure 3A). The ASDR of NDs decreased in all age groups across a 30-year time frame. The ASDR aged 1 to 4 decreased strongly in girls (−4.2% per year, from 7080.0 to 2133.4 per 100,000) and boys (−4.1% per year, from 6083.9 to 1926.2 per 100,000) (Figure 3B).

The ASIR attributed to protein-energy malnutrition in females increased fastest at age 1 to 4 (0.9% per year, from 5433.8 to 5340.7 per 100,000), followed by age 75+ (0.7% per year, from 1057.1 to 1533.4 per 100,000) and age 60 to 74 (0.7% per year, from 1043.1 to 1454.4 per 100,000). In males, ASIR increased significantly in age 60 to 74 (1.2% per year, from 1407.5 to 2081.4 per 100,000) and age over 75 (1.1% per year, from 1506.9 to 2194.5 per 100,000) (Appendix A). The ASIR of vitamin A deficiency and the ASDR of dietary iron deficiencies all strongly declined in the age group over 45 in both sexes (Appendix A).

### 3.4. Cohort Effect of NDs and Its High-Risk Subcategories with Increased Risk of Incidence in LMICs

In females, incidence rate of NDs increased 1.1% per successive 5-year birth cohort from 1995 to 1975. This increase accelerated to 2.4% per 5-year birth cohort from 1975 to 2005 and kept stable during 2005 to 2015. In males, incidence rates increased by 1.2% per 5-year birth cohort from 1910 to 1970, moderated to an increase of 0.6% from 1970 to 2005, and further declined 1.5% per 5-year birth cohort from 2005 to 2015 (Figure 4A).

The cohort effect on increased incidence risk of protein-energy malnutrition was generally consistent with overall NDs. In females, protein-energy malnutrition incidence increased 0.7% per successive 5-year birth cohort from 1895 to 1975 and the increase accelerated to 1.8% from 1975 to 2015. In males, protein-energy malnutrition incidence increased 1.2% per successive 5-year birth cohort from 1910 to 1970, moderated to an increase of 0.7% from 1970 to 2005, and further substantially declined 1.6% per successive 5-year cohorts born until 2015 (Figure 4B).

### 3.5. Spatial Patterns Differences of NDs Burden at National Level

Figure 5, Appendix A showed the tendency in sex-specific ASIR and ASDR of NDs and its main subcategories at national level. In order to classify countries by their evolving patterns in NDs burden, we performed a quadrant analysis using the sex-specific ASIR and ASDR in 2019 and their tendency during the study period. We calculated the 33rd and 66th percentiles (lower and upper terciles) in both measures based on statistical experience to describe the national distribution of NDs burden, and countries were classified into nine categories.

In females, the ASIR of NDs declined in 98 (73.1%) of 134 LMICs and increased in 18 (13.4%). Thirty-three (33.7%) of ninety-eight declined countries for females had an ASIR in the upper tercile of the distribution in 2019, representing countries with the highest ASIR in 2019. Furthermore, 10 of those 33 countries were in the upper tercile of the distribution of the EAPC for 1990–2019, representing the group with the smallest reduction in ASIR. In 18 countries with increased ASIR trend in female, six had the highest ASIR in 2019 located in the upper tercile of the distribution, and three of those six countries had the fastest increase: Turkey (1.4% per year, 1405.8 per 100,000 in 2019), Montenegro (0.9% per year, 1399.0 per 100,000 in 2019), and North Macedonia (0.6% per year, 1442.8 per 100,000 in 2019) (Figure 5A).

In males, 94 (70.1%) of 134 LMICs decreased, and 17 (12.7%) increased. Thirty-two of ninety-four decreased countries had the highest ASIR, and ten of these thirty-two countries had the smallest reduction. In 17 countries with increased ASIR trend in male, three of six countries located in the upper tercile of the distribution had the fastest increase: Bhutan (2.4% per year, 4489.4 per 100,000 in 2019), China (1.3% per year, 2482.3 per 100,000 in 2019), and Albania (1.1% per year, 1648.3 per 100,000 in 2019) (Figure 5B).

The only increased ASDR tendency attributed to NDs was observed in Zimbabwe females (1.9% per year, 1525.4 per 100,000 in 2019) and males (1.0% per year, 1710.9 per 100,000 in 2019); other countries all declined or were stable. Except for the ASIR of protein-energy malnutrition remaining increased in females of 21 countries and males of 18 countries, decreases or stable ASIR and ASDR were observed for all main subcategories in the majority of LMICs from 1990 to 2019 (Appendix A).

### 3.6. The Relationship between NDs Burden and Gender Development Index

Appendix A present the relationship between the NDs and its main subcategories’ burden by sex with Gender Development Index in LMICs in 2019. Overall, the ASIR and ASDR by sexes due to overall NDs were all negatively correlated with GDI (*p* < 0.05). The high ASIR and ASDR due to overall NDs generally occurred in countries with low sex gap levels and so did results in main subcategories in both sexes (*p* < 0.05).

## 4. Discussion

We systematically compared and evaluated the burden of NDs and its main subcategories at overall and national levels in LMICs. Our study suggested that LMICs contributed an increasingly larger share of NDs worldwide from 1990 to 2019. Although no trend was evident for overall females during the 30-year period, females in the age group 15+ born before 2005 experienced an increased age-standardized incidence of NDs. Males were observed to have significantly increased tendencies of incidence in overall NDs and its subcategory, protein-energy malnutrition, between 1990 and 2019, particularly in the age group 45+ born before 1970.

The results showed that NDs in LMICs were still a key area for global nutrition governance. Several factors may contribute to higher NDs in LMICs. Environmental degradation, climatic variability, economic downturns, population pressure, and the COVID-19 pandemic have undercut efforts to end hunger, food insecurity, and malnutrition [1], and the food supply in many countries remains far below energy requirements [18]. Unemployment and falling incomes, in addition to high prices for nutritious foods, have increased the pressure on groups with a low socioeconomic status [19,20] who have made changes to their diet to rely more on single staple foods [21]. Rapid urbanization has caused changes in dietary patterns, which shifted toward energy-dense but low-nutrient fast foods, causing the double burden of malnutrition and overweight [22]. The use of contaminated water due to climate change for the cleaning and processing of food has contributed to foodborne diseases [22]. Moreover, the aging population exacerbates malnutrition-related diseases [20]. Unhealthy lifestyle and disease, smoking, alcohol, drugs, and AIDS accelerated nutrient loss [23,24,25].

The time-stable trends that we observed for females and young children in incidence need to be interpreted with caution. Firstly, the trends may be an artifact caused by increased attention to maternal and child health work and improved health awareness among females in LIMCs. Higher incidence risk for females born in the 1975 to 2015 cohort may also be related to more frequent health monitoring. In addition, the lack of a uniform definition of NDs and heterogeneous nutrition screening practices presents great challenges for the accurate estimation of the extent of NDs before malnutrition diagnosis consensus published by the European Society for the Parenteral and Enteral Nutrition (ESPEN) in 2018 [26]. The terms “malnutrition risk” and “malnutrition” and the tools “malnutrition screening” and “risk screening” have been used interchangeably, which has caused further inconsistencies in assessment results [27,28,29]. Among micronutrient deficiency measurement tools, dietary intake data are prone to underreporting [27]; proxy indicators do not fully reflect micronutrient status [28]; there is a lack of consensus on the best biomarkers and cutoff values; and results can be influenced by inflammation and infection, the subject’s hydration status, age and sexes, kidney function, the analytical method used, and the season or time of annual dietary shortages [29].

The observation that time trends were more evident in males than in female may support a genuine trend in incidence for NDs and its subcategory, protein-energy malnutrition, and the trend was driven primarily by 45+ born before 1970. The aggravating trend of the aging population impacts nutritional deficiency in older male from physical and physiological impairments, as well as psychosocial influences. Aging is usually accompanied with reduced lean body mass in physical examination, resulting in a reduced metabolic rate and a decline in total energy requirements [30]. Changes to the gastrointestinal tract and sensory function caused poor appetite and reduced food intake [31,32]. The high prevalence of chronic diseases in older individuals often result in a loss of dexterity, coordination, mobility, and difficulty with food preparation [33]. Medication and hospitalization often bring the consequence of altered absorption, utilization, or excretion of essential nutrients [30]. In addition, financial restraints and living alone among elders reduced food security and resulted in inappropriate food choices [31].

Exploration of ND burden data and change patterns at the national level is useful for analyzing social inequality between countries [34]. We analyzed inequality between countries from bivariate (burden in 2019 and changes), and Bhutan’s male population has observed the top threat of nutritional deficiencies burden. Although Bhutanese male consistently reported higher consumption of fruit and vegetables than women, their level of consumption still does not meet WHO recommendations. Moreover, Bhutanese men use tobacco and alcohol more frequently and are more physically active than Bhutanese women [19]. Further multivariate sub-national analyses and confirmation of this inequality are needed. We also analyzed the causal inference between sex-specific burden in 2019 with gender development. Countries with small sex differences had less burden of NDs in both females and males. Equal sex power means equal access to food, distribution, and access to medical resources.

Gratifyingly, over the past 30 years, the ASDR of NDs and subcategories decreased significantly. This may have appeared due to improvements in the healthcare system, implementation of infant nutrition intervention program [35,36], increased breast-feeding rates, better complementary feeding [37], increased immunization coverage, improved water and sanitation measures, and food fortification measures in LMICs.

Notably, substantial discrepancies in the NDs burden have persisted between sex, age, subcategories, and spatial patterns in LMICs, despite a relative decrease in ND burden. It is suggested that the focus of prevention and treatment of nutritional deficiencies in LMICs is still women, children, and elderly males. Addressing the burden of nutritional deficiencies in vulnerable populations requires macro political commitment, economic capital, cultural capital, and social capital. Nutritional supplementation and food fortification most effectively resolve nutritional deficits in a cost-effective manner, and effective measures to address nutritional deficiencies are recommended by international organizations [38]. However, these approaches have not solved the problem to the desired extent because of poor governmental support, supervision, and low compliance to legislation. It will be crucial to not rely on supplementation and food fortification strategies alone, particularly considering country and regional contexts and food consumption patterns, as well as potential food vehicles among vulnerable populations in need.

To the best of our knowledge, this study was the first comprehensive overview and exploration of the ND epidemic, DALYs burden, type differences, and their evolving pattern stratified by sex and age in LMICs. This study had limitations that must be considered. Although GBD 2019 adds biased adjustments to allow for low-quality sampling, survey methods, and other methodological deficiencies in data sources, the accuracy of the results for NDs largely depends on the quality and quantity of the data input into the models [13,39,40]. In addition, the lack of a gold standard for diagnosis and the interchangeable use of the terms may underestimate the scale of NDs.

## 5. Conclusions

Our results suggested the spread of the epidemic of NDs in age 15+ born before 2005, especially in males, and the DALYs burden was ebbing per age group. In the main subcategories of NDs, protein-energy malnutrition had an increased risk in males aged 45+ born before 1970; the ASIR of vitamin A deficiency and the ASDR of dietary iron deficiency experienced decreases over three decades, and they remained at the heaviest level in 2019, especially in children under 5 years. The heaviest burden of NDs occurred in females from Turkey, Montenegro, and North Macedonia and males from Bhutan, China, and Albania.

## Figures and Tables

**Figure 1 nutrients-14-00931-f001:**
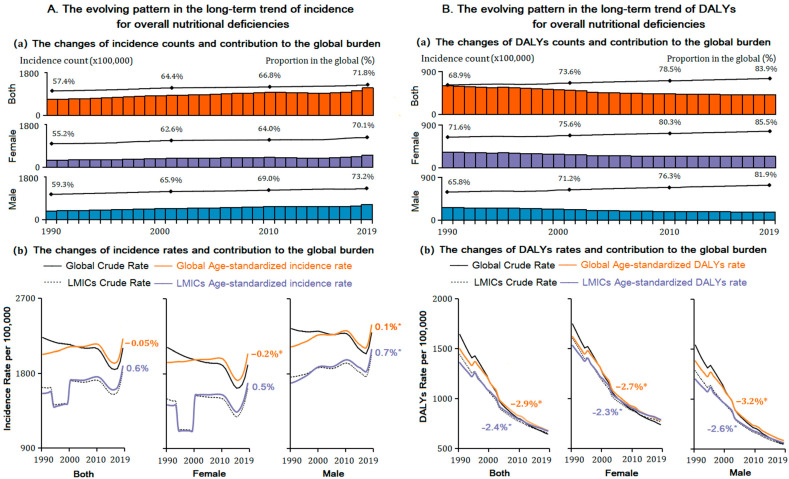
The sex-specific incidence and DALYs counts, age-standardized rate, and their trend for nutritional deficiencies (NDs) in LMICs from 1990 to 2019, and their contribution to the global burden. (**A**) Incidence; (**B**) DALYs. DALYs = disability-adjusted life years. *Note: (*) Indicates statistically significant trend (p < 0.05).*

**Figure 2 nutrients-14-00931-f002:**
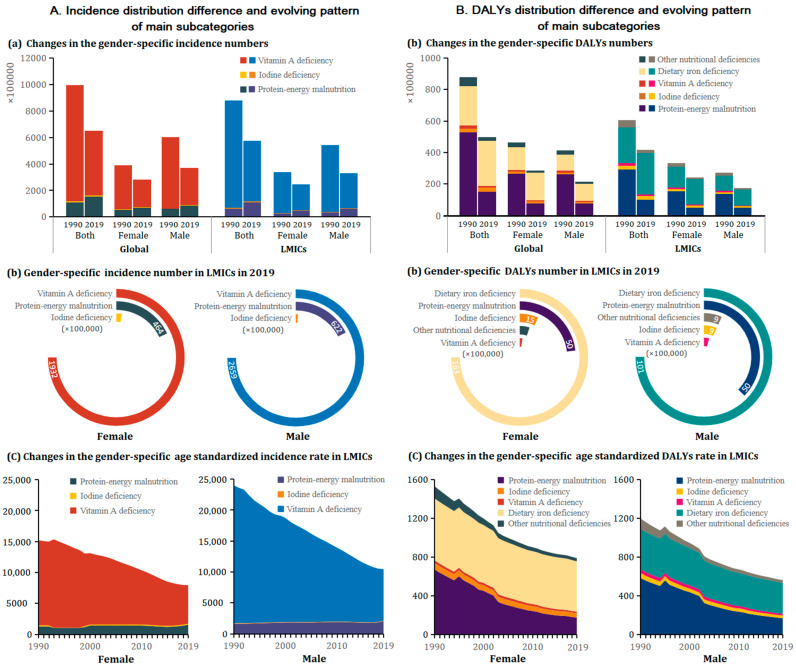
The sex-specific incidence and DALYs counts and age-standardized rate for main subcaegories in LMICs from 1990 to 2019, and their contribution to the global burden. (**A**) Incidence; (**B**) DALYs. DALYs = disability-adjusted life years.

**Figure 3 nutrients-14-00931-f003:**
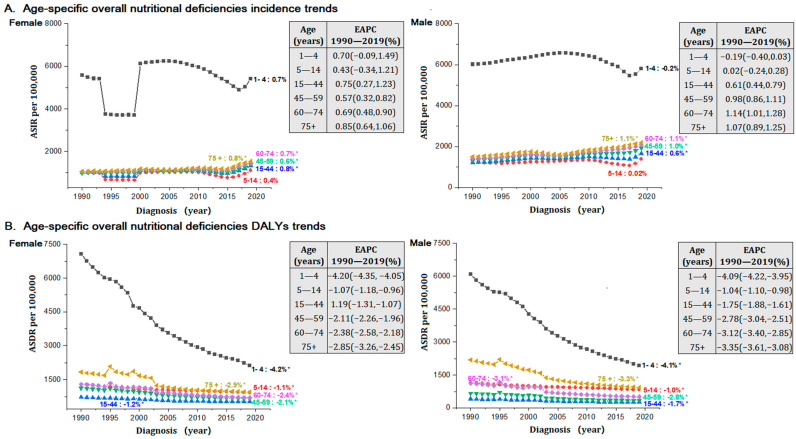
The age-sex specific trends of age-standardized incidence rate and age-standardized DALYs rate for nutritional deficiencies (NDs) in LMICs from 1990 to 2019. (**A**) Age-standardized incidence rate (ASIR); (**B**) age-standardized DALYs rate (ASDR). DALYs, disability-adjusted life years; EAPC, estimated annual percent change. *Note: (*) Indicates statistically significant trend (p < 0.05).*

**Figure 4 nutrients-14-00931-f004:**
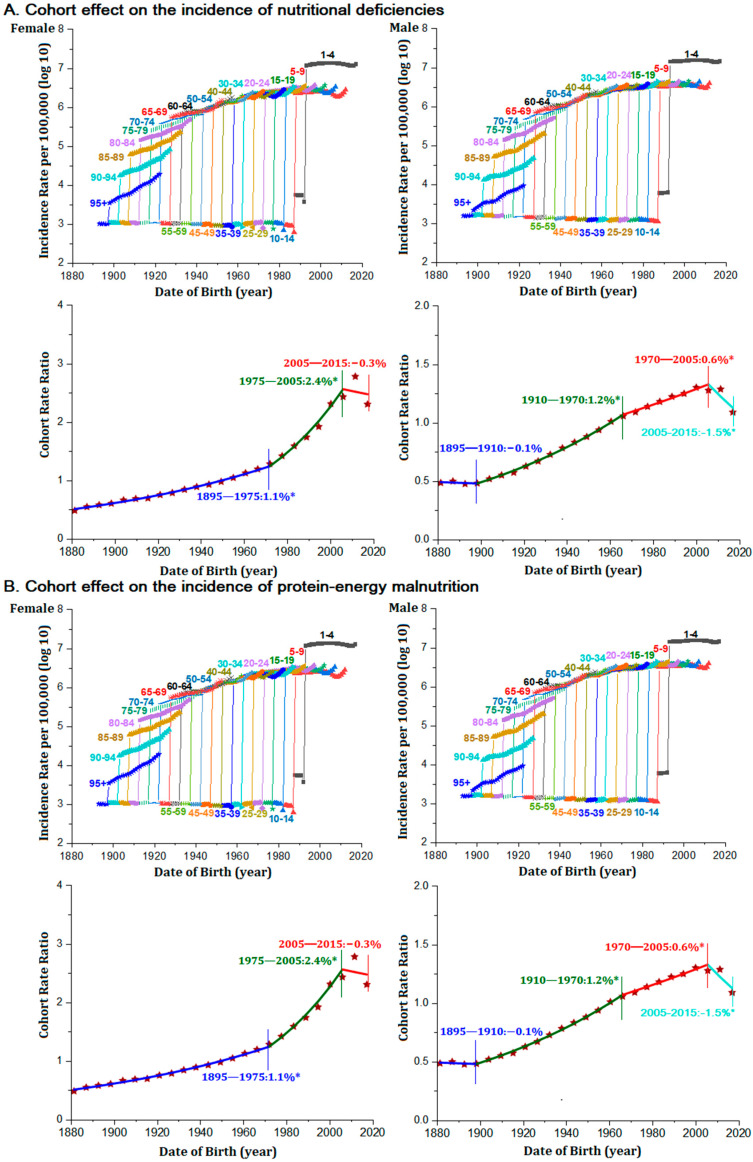
Age-sex specific incidence rates with increased risk across birth cohort and annual percent change of cohort rate ratios in LMICs for nutritional deficiencies and its subcategories. (**A**) Overall nutritional deficiencies. (**B**) Subcategory protein-energy malnutrition. *Note: (*) Indicates statistically significant change in each successive birth cohort (p < 0.05).*

**Figure 5 nutrients-14-00931-f005:**
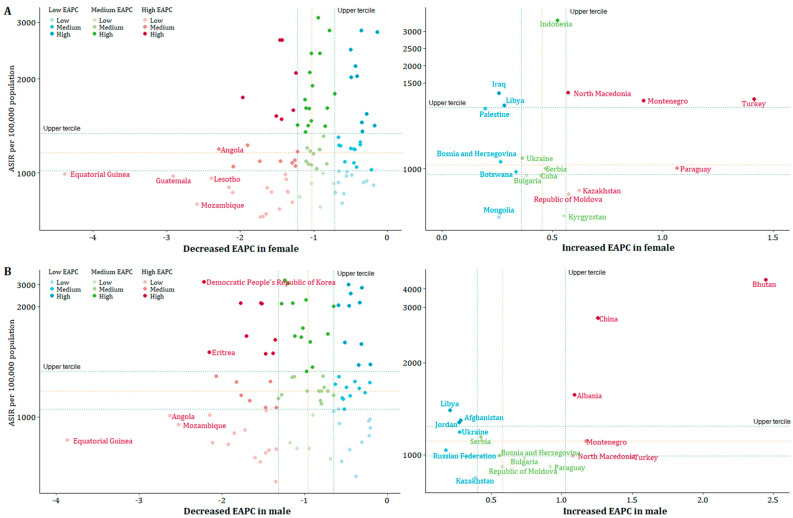
Age-standardized incidence rate in 2019 and its estimated annual percentage change in 1990–2019 due to nutritional deficiencies for all ages stratified by sex at national levels in LMICs. (**A**) Female; (**B**) male. *Note: Dots represent countries, color-coded according to quadrants defined by lower and upper terciles (33 and 66 percentiles) of the distribution of countries according to the ASIR per 100,000 population and the estimated annual percentage change of the ASIR per 100,000 population in 1990–2019. The countries marked in the figure represent the top five countries with the fastest decline per year and all countries with increased trend.*

**Table 1 nutrients-14-00931-t001:** Sex-Specific incidence and DALYs numbers, age-standardized rate per 100,000 for nutritional deficiencies (NDs) and its main subcategories, and their trend from 1990 to 2019 in LMICs.

	Incidence				DALYs			
	No. in Thousands, 2019	No. Change, 1990–2019, %	ASIR per 100,000, 2019	ASIR. Change, 1990–2019, %	No. in Thousands, 2019	No. Change, 1990–2019, %	ASDR per 100,000, 2019	ASDR. Change, 1990–2019, %
Overall nutritional deficiencies								
Both	116,480.4	70.4%	1883.2	0.6 (−0.2, 1.4)	41,777.8	−30.9%	674.4	−2.4 (−2.6, −2.2)
Female	51,405.6	66.0%	1681.5	0.5 (−1.2, 2.1)	24,321.3	−26.8%	788.3	−2.3 (−2.4, −2.1)
Male	65,074.9	74.1%	2086.6	0.7 (0.6, 0.8)	17,456.5	−36.0%	562.1	−2.6 (−2.9, −2.3)
Main subcategories of nutritional deficiency								
Protein-energy malnutrition								
Both	108,645.94	78.2%	1202.48	0.8 (0.1, 1.4)	9988.4	−65.9%	171.7	−4.4 (−5.1, −3.7)
Female	46,439.45	73.9%	1526.16	0.6 (−0.5, 1.8)	4988.4	−68.0%	174.7	−4.6 (−5.1, −4.0)
Male	62,206.48	81.5%	2000.04	0.8 (0.8, 0.9)	5000.1	−63.6%	169.8	−4.3 (−5.4, −3.1)
Iodine deficiency								
Both	7834.5	6.3%	120.3	−0.6 (−2.5, 1.3)	2357.9	−2.4%	36.4	−1.6 (−1.8, −1.5)
Female	4966.1	16.5%	155.3	−0.3 (−3.0, 2.5)	1469.2	9.4%	45.5	−1.3 (−1.5, −1.2)
Male	2868.4	−7.8%	86.6	−1.1 (−1.2, −1.1)	888.7	−17.2%	27.4	−2.1 (−2.3, −2.0)
Vitamin A deficiency								
Both	459,155.8	−43.5%	7291.5	−3.1 (−3.2, −3.0)	1057.8	−38.8%	17.5	−2.2 (−2.4, −2.1)
Female	193,227.3	−36.9%	6238.7	−2.7 (−2.9, −2.5)	476.2	−31.6%	16.2	−1.9 (−2.1, −1.7)
Male	265,928.5	−47.5%	8325.3	−3.4 (−3.5, −3.2)	581.6	−43.7%	18.8	−2.5 (−2.5, −2.4)
Dietary iron deficiency								
Both	-	-	-	-	26,390.6	17.4%	417.4	−0.8 (−0.8, −0.8)
Female	-	-	-	-	16,327.7	24.3%	518.6	−0.7 (−0.8, −0.7)
Male	-	-	-	-	10,063.0	7.7%	316.5	−0.9 (−1.0, −0.9)
Other nutritional deficiencies								
Both	-	-	-	-	1983.1	−56.8%	31.6	−4.6 (−4.8, −4.3)
Female	-	-	-	-	1059.9	−57.3%	33.5	−4.5 (−4.6, −4.3)
Male	-	-	-	-	923.2	−56.1%	29.9	−4.5 (−4.9, −4.0)

Abbreviation: NDs, nutritional deficiencies; LMICs, low- and middle-income countries; ASIR, age-standardized incidence rate; ASDR, age-standardized DALYs rate; DALYs, disability-adjusted life years.

## Data Availability

The data underlying this article are available in the Global Health Data Exchange at http://ghdx.healthdata.org/ihme data (accessed on 6 January 2022).

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
