# Peer review of "Evolving Patterns of Nutritional Deficiencies Burden in Low- and Middle-Income Countries: Findings from the 2019 Global Burden of Disease Study"

_nutrients, 2022, doi:10.3390/nu14050931_

Round 1
Reviewer 1 Report
Review and opinion: Comments and suggestions
The study is quite interesting. I particularly appreciate works of this nature. The study problem is very important and also reveals the care that quality global health should prioritize. The methodology is well described and appropriately. Comparisons between nations are very important, especially nowadays due to the great social, economic and scholastic inequalities that prevail in a globalized world.
I understand, however, that the authors could take advantage of the richness and robustness of the data they have and try (see, this is just a suggestion) to establish a correlation analysis between the indicators of nutritional deficiencies (different types of nutrients) with economic indicators, such as such as GDP, per capita income, or scholastic indicators and even with varied social indicators. I understand that such analyzes could enrich the study to be published.
It is an Important work. Congratulations to the authors.
Author Response
Response:
Thank you for your affirmation and suggestion. In this manuscript, we use the Gender Development Index (GDI), a macro-indicator of low- and middle-income countries, to explore the relationship between sex-specific nutritional deficiency burden with the degree of sex development. In future research, we will continue to explore the economic, cultural and social factors associated with the burden of nutritional deficiencies.
The correlation analysis of Gender Development Index (GDI) with sex-specific burden is as follows:
Gender Development Index set
In this study, we explored the relationship between sex-specific NDs incidence and DALYs burden with Gender Development Index (GDI). The GDI is a direct measure of sex gap in human development achievements. The GDI is a direct measure of the sex gap in human development achievements by accounting for sex disparities in health, knowledge, and living standards using the same methodology as in the Human Development Index (HDI). The GDI is the ratio of the HDIs calculated separately for female and male, and shows the female HDI as a percentage of the male HDI. The GDI score is closer to 1, indicating that the sex difference is smaller. The data was acquired from the United Nations Development Programme (UNDP; http://hdr.undp.org/en/data).
Statistical analysis
We finally analyzed the relations between sex-specific burden and Gender Development Index (GDI) using the Pearson correlation analysis, and all analyse using SAS 9.4 statistical software (SAS Institute Inc., Cary, NC, USA) and R version 3.3.
The relationship between NDs burden and Gender Development Index
eFig. 6-7 in the supplement present the relationship between the NDs and its main subcategories burden by sex with Gender Development Index in LMICs in 2019. Overall, the ASIR and ASDR by sexes due to overall NDs were all negatively correlated with GDI (P<0.05). The high ASIR and ASDR due to overall NDs generally occurred in the countries with low sexes gap levels. So did results in main subcategories in both sexes (P<0.05).
Fiure 6: The correlation between sex-specific age-standardized incidence rates (ASIR) and age-standardized DALYs rates (ASDR) for nutritional deficiencies (NDs) with Gender development index (GDI) in low-and middle-income countries (LMICs) in 2019. A. Female; B. Male.
Fiure 7: The correlation between sex-specific age-standardized incidence rates (ASIR) and age-standardized DALYs rates (ASDR) for subcategories of nutritional deficiencies with Gender development index (GDI) in 2019. A. Age-standardized incidence rates (ASIR); B. Age-standardized DALYs rates (ASDR). DALYs, disability-adjusted life years.

Reviewer 2 Report
The article makes an important contribution to public health by studying the indicators of incidence and disability-adjusted life years
(DALYs) to measure the burden of NDs and its main subcategories in LMICs, including proteinenergy malnutrition, iodine deficiency, vitamin A deficiency, dietary iron deficiency, and other nutritional deficiencies, by sex, age and spatial patterns .
The discussion needs to be improved, especially in the discussion about nutrient supplementation/fortification. I suggest inserting the discussion into a greater risk of nutritional deficiencies, such as pregnant women, children and the elderly.
I recommend adjustments in the quality of the figures to make them clearer to the reader. Text also needs formatting.
Author Response
The article makes an important contribution to public health by studying the indicators of incidence and disability-adjusted life years
(DALYs) to measure the burden of NDs and its main subcategories in LMICs, including proteinenergy malnutrition, iodine deficiency, vitamin A deficiency, dietary iron deficiency, and other nutritional deficiencies, by sex, age and spatial patterns .
The discussion needs to be improved, especially in the discussion about nutrient supplementation/fortification. I suggest inserting the discussion into a greater risk of nutritional deficiencies, such as pregnant women, children and the elderly.
I recommend adjustments in the quality of the figures to make them clearer to the reader. Text also needs formatting.
Response 1:
Thank for your suggestion. We intensify discussions on nutritional supplementation and fortification for vulnerable groups in our discussions. The details are as follows:
Notably, substantial discrepancies in the NDs burden have persisted between sex, age, subcategories and spatial patterns in LMICs, despite a relative decrease in the NDs burden. It is suggested that the focus of prevention and treatment of nutritional deficiencies in LMICs is still women, children, male elderly. Addressing the burden of nutritional deficiencies in vulnerable populations requires macro political commitment, economic capital, cultural capital, and social capital. Nutritional supplementation and food fortification are most effectively resolve nutritional deficits in a cost-effective manner effective measures to address nutritional deficiencies recommended by international organizations. 40 However, these approaches have not solved the problem to the desired extent because of poor governmental support, supervision and low compliance to legislation. It will be crucial not to rely on supplementation and food fortification strategies alone, particularly consider country and regional contexts where food consumption patterns, potential food vehicles among vulnerable populations in need.
At the same time, we have re-adjusted the clarity of the figures.
References
- Vosti SA, Kagin J, Engle-Stone R, Brown KH. An economic optimization model for improving the efficiency of vitamin A interventions: an application to young children in Cameroon. Food Nutr Bull 2015;36(3 Suppl):S193–